# Aberrantly Glycosylated IgA1 in IgA Nephropathy: What We Know and What We Don’t Know

**DOI:** 10.3390/jcm10163467

**Published:** 2021-08-05

**Authors:** Yukako Ohyama, Matthew B. Renfrow, Jan Novak, Kazuo Takahashi

**Affiliations:** 1Department of Biomedical Molecular Sciences, Fujita Health University School of Medicine, Toyoake, Aichi 470-1192, Japan; yukako.oyama@fujita-hu.ac.jp; 2Departments of Biochemistry and Molecular Genetics and Microbiology, University of Alabama at Birmingham, Birmingham, AL 35294, USA; renfrow@uab.edu (M.B.R.); jannovak@uab.edu (J.N.)

**Keywords:** IgA nephropathy, aberrantly glycosylated IgA1, galactose-deficient IgA1, glycosylation of IgA1, biomarker

## Abstract

IgA nephropathy (IgAN), the most common primary glomerular disease worldwide, is characterized by glomerular deposition of IgA1-containing immune complexes. The IgA1 hinge region (HR) has up to six clustered *O*-glycans consisting of Ser/Thr-linked *N*-acetylgalactosamine usually with β1,3-linked galactose and variable sialylation. Circulating levels of IgA1 with abnormally *O*-glycosylated HR, termed galactose-deficient IgA1 (Gd-IgA1), are increased in patients with IgAN. Current evidence suggests that IgAN is induced by multiple sequential pathogenic steps, and production of aberrantly glycosylated IgA1 is considered the initial step. Thus, the mechanisms of biosynthesis of aberrantly glycosylated IgA1 and the involvement of aberrant glycoforms of IgA1 in disease development have been studied. Furthermore, Gd-IgA1 represents an attractive biomarker for IgAN, and its clinical significance is still being evaluated. To elucidate the pathogenesis of IgAN, it is important to deconvolute the biosynthetic origins of Gd-IgA1 and characterize the pathogenic IgA1 HR *O*-glycoform(s), including the glycan structures and their sites of attachment. These efforts will likely lead to development of new biomarkers. Here, we review the IgA1 HR *O*-glycosylation in general and the role of aberrantly glycosylated IgA1 in the pathogenesis of IgAN in particular.

## 1. Introduction

IgA nephropathy (IgAN) is a mesangioproliferative glomerulonephritis associated with depositions of IgA-containing immune complexes [1]. These deposits are exclusively of IgA1 subclass [2]. Recurrence of IgAN is common in patients after transplantation [3,4,5,6]. Conversely, when kidneys with subclinical IgA deposits are transplanted into patients with end-stage renal disease other than IgAN, the deposited IgA in the donor kidney is cleared [7,8,9]. Furthermore, IgA deposits disappear when bone-marrow transplantations are performed for patients with IgAN [10]. Conversely, IgAN can develop after bone-marrow transplantation due to a non-graft-versus-host-disease-related multi-hit process associated with glomerular deposition of aberrantly glycosylated IgA1 (see for details below) [11]. Based on these observations, it is thought that the glomerular IgA deposits are derived from circulating IgA1-containing complexes. Elevated IgA serum levels alone are not sufficient to induce IgAN, as patients with IgA myeloma, who have abnormally increased serum IgA levels, rarely suffer from concomitant IgAN [12,13,14]. According to these observations, the pathogenetic mechanisms underlying IgAN appear to involve qualitative features of IgA1 rather than a mere elevation in blood levels of total IgA1.

Patients with IgAN present with a variety of clinical, laboratory, and histopathological findings. In addition to primary IgAN, many patients with liver disease, collagen disease, inflammatory bowel syndrome, malignant tumor, and infection exhibit IgA glomerular deposition, i.e., secondary IgAN [15,16]. Due to these many clinical features and types of secondary IgAN, it has been assumed that IgAN may represent a group of diseases rather than a single disease [17,18].

Most human circulatory IgA is derived from B cells in the bone marrow. Ninety-percent of IgA1 is mainly present as a monomeric form [19], whereas glomerular IgA deposits contain mainly polymeric IgA1, i.e., two or more monomers connected by joining chain (J chain) [20,21]. IgA deposition may not necessarily be associated with glomerular damage, unless it is accompanied by deposition of complement [22,23]. The development of IgAN likely involves several complex mechanisms, including (over)production and glomerular deposition of IgA1-containing immune complexes, activation of mesangial cells and subsequent glomerular injury [14,15,18,24,25,26,27]. As many patients with IgAN exhibit exacerbation concurrently with upper-respiratory-tract infections, the involvement of mucosal immunity in the disease pathogenesis is suspected [25,28].

IgA1 is a highly glycosylated molecule, carrying both *N*-linked and *O*-linked glycans. The importance of aberrant IgA1 *O*-glycosylation in the pathogenesis of IgAN has been first observed based on differences in lectin reactivities with serum IgA1 from patients with IgAN vs. healthy subjects [29]. This conclusion was further supported by the observed enrichment of aberrant IgA1 *O*-glycoforms in the glomerular immunodeposits of IgAN patients [30,31].

The circulating levels of polymeric IgA with aberrantly glycosylated IgA1, with some *O*-glycans in the hinge region (HR) galactose (Gal)-deficient, are elevated in most patients with IgAN [32,33]. Mesangial deposition of macromolecular IgA1 is thought to occur in patients with IgAN due to the formation of immune complexes. Particularly, autoantibodies specific for Gd-IgA1 bind Gd-IgA1 and form circulating immune complexes (CICs) [25]. Furthermore, IgA and aberrantly glycosylated IgA1 may bind to the soluble form of the Fcα receptor (sCD89) [34], and aberrantly glycosylated IgA1 may form protein aggregates [35]. IgA1-contaning immune complexes activate the complement system, as evidenced by complement C3 in glomerular immunodeposits, leading to glomerular and renal tubular injury [36], as well as by the presence of C3 in the circulating IgA1-containing immune complexes (for review and details see [37,38,39,40,41,42]). Here, we review pathological and clinical significance of aberrant glycosylation of IgA1 in IgAN.

## 2. Structure of IgA

IgA production in humans is approximately 66 mg/kg/day, which is approximately twice that of IgG and 10 times that of IgM [43]. Two-thirds of IgA is secreted into the mucosal surfaces as the dominant isotype of secretory antibodies. Human IgA exists in two subclasses, IgA1 and IgA2; these two glycoproteins differ in number of *N*-glycans and the presence of *O*-glycans in IgA1. Compared with IgA2, IgA1 has clustered *O*-glycans in its HR, whereas IgA2 HR is short and without glycosylation sites. Figure 1a shows the glycan binding sites. The J chain, which is essential for the formation of polymeric IgA, has an *N*-glycan (Figure 1b). Most IgA in the mucosal tissues is produced by B cells as dimeric IgA with a disulfide-bond connected 16-kDa J chain. Polymeric IgA can bind to the polymeric immunoglobulin receptor (pIgR). Once polymeric IgA is bound to the pIgR at the basolateral side of the epithelium, it is transported to the luminal side through transcytosis, and a portion of pIgR is cleaved and remains attached to IgA to form secretory IgA (sIgA). This pIgR fragment, termed the secretory component (SC), has seven *N*-glycans (Figure 1c). Glycans on IgA and SC can bind or repel glycans on bacterial surfaces to prevent invasion into the blood/tissues [44].

### 2.1. Glycosylation of IgA1

Figure 2 shows the structure of IgA1 [46]. Clustered *O*-glycans on IgA1 HR are present only in hominid primates [47,48,49]. Human IgA1 has two *N*-glycosylation sites in the CH2 region and in the tailpiece (asparagine (Asn)^263^ and Asn^459^) (Figure 1a), and nine potential *O*-glycosylation sites (serine (Ser) and threonine (Thr)) in the proline (Pro)-rich HR (Figure 2a). Glycosylation in IgA1 HR occurs at specific sites [50,51,52,53]; usually three to six *O*-glycans are attached per HR [46,54], and the *O*-glycans of IgA1 are the core 1 glycans. Six different types of *O*-glycans can be found in IgA1 HR (Figure 2b). These features, number of attached *O*-glycans, glycan structures, and their attachment sites, contribute to the IgA1 HR *O*-glycoform diversity.

*O*-glycans of IgA1 are synthesized in a step-wise manner by glycosyltransferases in the Golgi apparatus of IgA1-secreting cells [56] (Figure 3). IgA1 HR *O*-glycosylation is initiated by the attachment of *N*-acetylgalactosamine (GalNAc) to Ser or Thr, catalyzed by UDP-*N*-acetylgalactosaminyltransferase 2 (GalNAc-T2) [51]. Other GalNAc-Ts can also participate in IgA1 HR glycosylation [57,58,59,60]. The addition of galactose (Gal) is mediated by the core 1 β1,3-galactosyltransferase (C1GalT1). The stability of C1GalT1 protein depends on its interaction with a specific molecular chaperone, Cosmc. In the absence of Cosmc, the C1GalT1 protein is rapidly degraded and, thus, Gal cannot be attached to GalNAc [61]. Furthermore, *N*-acetylneuraminic acid (NeuAc) is transferred to Gal and/or GalNAc by α2,3-sialyltransferase (ST3Gal1) and α2,6-sialyltransferase (ST6GalNAc2), respectively. If NeuAc is attached to GalNAc prior to the attachment of Gal, this premature sialylation precludes subsequent attachment of a Gal residue [62,63,64].

### 2.2. Biological Roles of O-Glycosylation in IgA1 HR

Protein glycosylation impacts many biological functions [65], and *O*-glycosylation of IgA1 HR also influences the structural and functional characteristics of the IgA1 molecule. *O*-glycosylation in IgA1 HR probably imparts the T-shaped antigen-binding fragment (Fab) arms. The difference in the arrangement of Fab and Fc (fragment crystallizable) segments between IgA1 and IgA2 was previously demonstrated using X-ray and neutron scattering procedures [66]. The structural differences between IgA1 and IgA2 may be driven by the backbone amino-acid sequence and *O*-glycosylation of IgA HR; *O*-glycosylation of Ser/Thr residues affects *cis/trans* isomerization of Pro residues in HR [67]. Appropriate glycosylation of HR may decrease unfavorable relative orientations between Fab and Fc in IgA1 by stabilizing the conformation of HR. In addition, HR *O*-glycans add hydrophilic characteristics to IgA1 [55].

In addition to the structural influence on IgA1 molecules, IgA1 HR *O*-glycans mediate antigen-nonspecific binding to bacteria [58,68]. This function enables IgA to participate in innate immunity, in addition to adaptive immunity. It is possible that the *O*-glycans in IgA1 HR are related to enhanced innate immune functions of hominoid-primates IgA1 subclass.

### 2.3. Aberrantly Glycosylated IgA1

IgA1 with some HR *O*-glycan(s) Gal-deficient can be detected by lectins, such as *Helix aspersa* agglutinin (HAA), a lectin specific for terminal GalNAc [32,69,70,71,72]. HAA in enzyme-linked immunosorbent assay (ELISA) revealed elevated levels of Gal-deficient IgA1 (Gd-IgA1) in the circulation in patients with IgAN. The term “Gd-IgA1” is often used as a synonym for “aberrantly glycosylated IgA1”, but it should be noted that not all *O*-glycans in Gd-IgA1 HR have to be Gal-deficient. Other types of aberrant glycosylation of IgA1 may include underglycosylation, i.e., a lower number of *O*-glycans per HR compared to IgA1 from healthy subjects, or over-sialylation, elevated amount of NeuAc in IgA1 HR in patients with IgAN [73,74,75,76]. It is to be noted that IgA1 *O*-glycans are presented as a heterogeneous mixture of glycoforms in patients with IgAN as well as in healthy subjects. Thus, “aberrantly glycosylated IgA1” is also present in the circulation of healthy individuals [53,77] and often a cut-off point is used to define “normal” vs. “abnormal” glycoforms of IgA1. To identify IgA1 *O*-glycoforms that are elevated in the circulation of patients with IgAN, quantitative analyses are required, including site-specific attachment of Gal-deficient *O*-glycan. The detailed structures of aberrantly glycosylated IgA1 specific for IgAN have not been reported. The levels of Gd-IgA1 produced by cultured IgA1-producing cells from peripheral blood correlate with the donors’ serum levels of Gd-IgA1 measured by lectin ELISA. This observation suggests that aberrantly glycosylated IgA1 originates due to the abnormal biosynthetic processes in IgA1-producing cells rather than a removal of Gal from circulatory IgA1 in the blood [76]. Reduced C1GalT1 activity and elevated ST6GalNAc2 activity in IgA1-producing cell lines are associated with production of Gd-IgA1 in IgAN [76].

## 3. Pathogenic Significance of Aberrantly Glycosylated IgA1

In IgAN, circulatory IgA1 with some glycans deficient in Gal is considered a disease-specific abnormality as this type of aberrant glycosylation is not observed for other glycoproteins with *O*-glycans, such as IgD or C1 inhibitor [78]. However, a high level of circulatory aberrantly glycosylated IgA1 alone does not induce glomerular injury [79]. Additional factors are thought to be involved in the development of IgAN, namely unique autoantibodies specific for Gd-IgA1 that enable formation of IgA1-containing immune complexes in the circulation, some of which deposit in the kidneys and induce glomerular injury [80,81]. Here, we discuss the factors related to the production of aberrantly glycosylated IgA1 (genetic factors, mucosal immunity) and Gd-IgA1 involvement in this postulated multi-hit mechanism.

### 3.1. Genetic Factors Associated to Gd-IgA1 Production

Increased serum levels of Gd-IgA1 are observed in 47% of first-degree relatives of patients with familial IgAN and 25% of first-degree relatives of patients with sporadic IgAN [79]. Furthermore, increased serum levels of aberrantly glycosylated IgA1 are observed in first-degree relatives of both pediatric and adult patients with IgAN [79,82,83,84,85], indicating a relationship between genetic background and aberrantly glycosylated IgA1. Regarding the production of aberrantly glycosylated IgA1, several single-nucleotide polymorphisms of *C1GALT1* [86,87,88] and *ST6GALNAC2* [88,89] genes represent IgAN-susceptibility alleles. However, no mutations were found in the exon of the gene encoding Cosmc, the molecular chaperone specific for C1GalT1 [90].

Genome-wide association studies of IgAN have identified many IgAN susceptibility loci, mainly in the MHC region at 6p21 [91,92], the *DEFA* locus at 8p23 [91,92,93], the *TNFSF13* locus at 17p13 [91], the *HORMAD2* locus at 22q12 [94], the *CFH/CFHR* locus at 1q32, the *ITGAM-ITGAX* locus at 16p11 [92], the *VAV3* locus at 1p13 [92] and the *CARD9* locus at 9q34 [92]. Except *C1GALT1*, genomic regions encoding other enzymes involved in IgA1 *O*-glycosylation have not been confirmed as IgAN susceptibility loci. Two genome-wide association studies confirmed that Gd-IgA1 serum levels are heritable and influenced by genetic variation at the *C1GALT1* gene [95,96]. Furthermore, two significant loci, in *C1GALT1* (rs13226913) and *C1GALT1C1* (rs5910940), which encode C1GalT1 and Cosmc, respectively, were identified in quantitative trait genome-wide association studies for serum levels of Gd-IgA1. In particular, the Gd-IgA1-increasing allele rs13226913 is rare or absent in some Asian populations and is the predominant allele in Europeans [96].

In addition, some microRNAs (miRNAs) are involved in regulating expression of some glycosyltransferases [97]. Relevant to IgAN, it was noted that expression of two miRNAs was associated with IgA1 *O*-glycosylation; expression of miR-148b targeting *C1GALT1* and miR-let-7b targeting *GALNT2* is elevated in peripheral blood mononuclear cells (PBMCs) of IgAN patients vs. healthy subjects [98,99]. These two miRNAs can be also detected in serum samples. A 2016 study found differences in the expression patterns between Caucasians and Asians, with significant upregulation in Caucasians [100]. From these reports, Gd-IgA1 levels are affected by the different alleles of the gene encoding *C1GALT1* and the different expression levels of miRNAs that control the expression of *C1GALT1* and *GALNT2*. Furthermore, the frequency of Gd-IgA1-increasing alleles and miR-148b that regulate *C1GALT1* expression are both higher in Caucasians than in Asians and may contribute to higher Gd-IgA1 levels in Caucasians than in Asians [95]. More recent studies identified additional mechanisms involving competing endogenous RNA molecules involved in regulation of IgA1 *O*-glycosylation, indicating that there is more to be discovered on the regulation of IgA1 *O*-glycosylation in general and in IgAN specifically [101,102,103]

### 3.2. Alteration of Mucosal Immunity Associated with Gd-IgA1 Production

Mucosal immunity is closely associated with IgAN. Patients with IgAN often present with an episode of macroscopic hematuria after upper-respiratory-tract or gastrointestinal tract infections. Furthermore, IgAN is associated with diseases involving mucosal abnormalities, such as ulcerative colitis, Crohn’s disease, and coeliac disease [104,105,106,107].

Mesangial IgA1 deposits in IgAN resemble mucosal IgA, wherein they are both polymeric and relatively poorly *O*-galactosylated [30,31,108,109]. As the number of polymeric IgA-secreting plasma cells is increased in the bone marrow of IgAN patients [110], it has been speculated that mis-homing of mucosally imprinted B cells to the bone marrow may mediate production of “mucosal IgA1” into the circulation. IgD *O*-glycosylation is apparently normal in patients with IgAN [78] and, thus, factors specific for IgA1-producing cells are thought to affect *O*-glycosylation of IgA1 in IgAN, such as the microenvironment with antigen-mediated activation and class-switching signals [111].

The tonsils represent a mucosa-associated lymphoid tissue involved in IgA production, and tonsillectomy, often combined with steroid pulse, has been reported to be an effective treatment of IgAN in east Asia [112,113,114]. With regards to the expression of *O*-glycosyltransferases in tonsillar CD19^+^ lymphocytes, *C1GALT1*, *C1GALT1C1*, and *GALNT2* have significantly reduced expression in IgAN patients compared to the subjects with chronic tonsillitis [115]. T-helper 2 (Th2)-cell polarity was observed in the tonsillar tissue of IgAN [116], and interleukin-4 (IL-4), a Th2 cytokine, reduced expression of C1GalT1 and Cosmc in a human B cell lines [117]. Studies using immortalized IgA1-secreting cell lines derived from peripheral blood of patients with IgAN and healthy controls provided additional insight into the regulation of glycogene expression by cytokines and Gd-IgA1 production [64,118]. IL-4 and, even more so, IL-6 enhance production of Gd-IgA1 due to further dysregulation of expression and activities of specific targets, including *C1GALT1, C1GALT1C1,* and *ST6GALNAC2* [64]. Leukemia inhibitory factor (LIF), an IL-6-related cytokine, also enhances Gd-IgA1 production [119]. Notably, both IL-6 and LIF enhance production of Gd-IgA1 in IgA1-secreting cell lines derived from IgAN patients due to abnormal signaling in JAK-STAT pathways that leads to further dysregulation of key glycosyltransferases [38,119,120,121,122]. Furthermore, a proliferation-inducing ligand (APRIL), known to play an important role in T cell-independent IgA class switching, is overproduced by CD19^+^ B cells in tonsil germinal centers in IgAN [118]. In addition, in IgAN-prone ddY mice and in the human B cell line, APRIL and IL-6 expression is upregulated via activation by Toll-like receptor 9 (TLR9) by CpG-oligonucleotides (CpG-ODN) [123], resulting in increased Gd-IgA1 production. In summary, altered cytokine production and/or signaling responses in IgA1-producing cells in mucosal tissues may be one of the reasons for over-production of aberrantly glycosylated IgA1.

### 3.3. Antigenicity of Gd-IgA1 Related to Autoantibody Production

IgG or IgA1 autoantibodies targeting aberrantly glycosylated IgA1 have been identified in patients with IgAN [69,124,125]. IgG autoantibodies are predominant [126] and are found in the glomerular deposits of all patients [80]. Furthermore, a correlation between serum levels of Gd-IgA1 and the corresponding IgG autoantibodies was observed in sera of patients with IgAN, but not in sera of disease or healthy controls [126]. As described above, the alterations of *O*-glycosylation in IgA1 HR likely generate conformational changes in IgA1, contributing to the exposure of epitopes for anti-Gd-IgA1 autoantibodies, and driving the formation of IgA1-containing immune complexes (Figure 4c). IgG produced from IgG-producing cell lines derived from IgAN patients is highly reactive with desialylated and degalactosylated IgA1 and specific alterations of the amino acid sequence in the complementarity-determining region 3 (CDR3) in the variable region of the IgG heavy chains impacts the reactivity with Gd-IgA1 [125]. Notably, serum levels of IgG autoantibodies specific for Gd-IgA1 correlate with proteinuria [125]. Although the mechanism that leads to the formation of these autoantibodies has not been elucidated, these antibodies may be produced against some viruses and Gram-positive bacteria that express GalNAc-containing structures on their surfaces and they may acquire cross-reactivity with Gal-deficient IgA1 [25,56].

### 3.4. Impact of IgA1 O-Glycosylation of Interactions with IgA Receptors

There are five well-known IgA receptors: FcαR1 (CD89), asialoglycoprotein receptor (ASGPR), polymeric Ig receptor (pIgR), transferrin receptor (TfR; CD71), and Fc α/μ receptor. Some of these receptors are involved in IgA removal from the circulation, either for catabolism or transcytosis to mucosal surfaces.

ASGPR, which is expressed on the sinusoidal/lateral aspect of hepatocytes, is a receptor mediating endocytosis and catabolism of glycoproteins, including IgA [128]. In patients with secondary IgAN due to liver cirrhosis, reduced clearance of IgA-containing immune complexes (IgA-IC) has been reported, consistent with aberrant expression of hepatic ASGPR [16]. In primary IgAN, large size of IgA-IC is thought to limit access through the fenestration of endothelial cells to the space of Disse and, thus, to the ASGPR on hepatocytes, resulting in extended circulation time of IgA-IC [111].

CD89, a membrane glycoprotein expressed by cells of myeloid lineage, such as neutrophils, monocytes, macrophages, and eosinophils, binds to the constant region of the heavy chain of both the monomeric and dimeric forms of IgA1 and IgA2. It is thought that CD89 has a role in the removal of IgA-antigen complexes from the circulation [129]. In patients with IgAN, membrane expression of CD89 on circulating myeloid cells is decreased [129] and binding of monoclonal IgA to CD89 is impaired [130], which may contribute to a delayed IgA clearance. Notably, CD89 *N*-glycans significantly modulate binding affinity to IgA [131]. In addition, increased levels of soluble CD89 (sCD89) complexed with IgA are found in some patients with IgAN (Figure 4d) [132]. IgA-IC with sCD89 is possibly produced by shedding its extracellular portion after binding with IC containing polymeric Gd-IgA1. In mice generated by backcrossing between α1-knock-in mice and human CD89 transgenic mice, which spontaneously express human IgA1 and CD89, a complete human IgAN phenotype develops [133]. In addition, IgA-sCD89 complex levels are significantly higher in pre-transplantation patients with IgAN than in control groups and are also found in the mesangial deposition of IgAN [134].

Regarding IgA receptors expressed in human mesangial cells, Fcα/μR and CD71 receptors were detected [135,136], whereas CD89, ASGP-R, and pIgR were absent [137]. CD71 binds polymeric IgA1, but not monomeric IgA1 or IgA2, co-localizes with mesangial IgA1 deposits, and is overexpressed in patients with IgAN [136,138]. In addition, aberrantly glycosylated IgA1 and polymeric IgA1 show higher affinity for CD71, indicating that the formation of Gd-IgA1 immune complexes contributes to mesangial TfR-IgA1 interaction and glomerular deposition in IgAN [139].

Additional receptors able to bind IgA1 were identified in human mesangial cells: integrin α1/β1 and α2/β1 and β1,4-galactosyltransferase 1 [140,141]. Expression of the latter receptor in glomeruli is increased in IgAN, indicating that β1,4-galactosyltransferase 1 may play a role in IgA clearance and in the initial response to IgA deposition [140,142]. This observation on the role of β1,4-galactosyltransferase was reported earlier for binding of IgA1 to various cells [143].

### 3.5. Complement Activation

In IgAN, glomerular co-deposition of IgA and complement components of C3 are commonly observed, being present in at least 90% of renal biopsies [144]. As the membrane attack complex, the terminal product of the complement activation, is observed at high frequency, activation of the complement pathway has been considered to play a major role in glomerular injury in IgAN [145]. Although markers of the classical pathway, such as C1q and C4, are rarely observed, co-deposition of C3b, factor P, properdin, and factor H (FH) are observed in almost 100%, 75–100%, and 30–90% of cases, respectively [39,145]. In addition, cases with deposition of C4d, mannose-binding lectin (MBL), ficolin, and MBL-associated serine protease were observed, and associations of MBL and L-ficolin deposition with severity of renal histological damage have also been reported [146]. These aspects suggest that the alternative and lectin pathways are involved in IgAN pathophysiology [37].

Although the impact of aberrant IgA1 glycosylation on complement activation is not clearly understood, some reports suggest a relationship between complement activity and IgA1 glycosylation abnormalities. Polymeric IgA is more efficient in binding C3 and inducing glomerular injury compared to monomeric IgA in rats [147,148]. In humans, polymeric IgA1 is highly reactive with GalNAc-specific lectin HAA and shows a higher binding capacity for MBL via C-type lectin and higher C4 activation ability [149]. As recent studies showed that MBL binds to *N*-glycans of IgG and IgM [150,151], *N*-glycosylation of polymeric IgA may thus be associated with MBL binding and lectin pathway activation. In the comparison of *N*-glycosylation of IgA heavy chains between monomeric and polymeric IgA, oligomannose structure is elevated in polymeric IgA [149]. However, the *N*-glycans of the heavily glycosylated secretory component may also contribute to complement activation, as related to elevated concentration of secretory IgA in polymeric IgA is observed in IgAN [149]. Further investigations are required to determine which components and structures of *N*-glycans are involved in lectin-pathway activation and whether *N*- or *O*-glycosylation of IgA1 are involved in the alternative pathways of complement activation.

## 4. Clinical Significance of Aberrantly Glycosylated IgA1

In recent years, the validity of Gd-IgA1 and its associated molecules as diagnostic and prognostic markers has been assessed. Serum Gd-IgA1 levels can now be quantified using ELISA. Furthermore, detection and quantification of IgA1 HR *O*-glycosylation varieties and analysis of *O*-glycan attachment sites have been performed using high-resolution mass spectrometry.

### 4.1. Approaches for Detection of Aberrantly Glycosylated IgA1

For the detection of galactose-deficient *O*-glycan in IgA1 HR, some lectins that specifically bind to GalNAc residue have been used, such as *Helix aspersa* (HAA) or *H. pomatia* agglutinin (HPA) [70] and *Vicia villosa* lectin (VVL) [29]. A lectin-based assay for the detection of Gd-IgA1 was developed [70], and we confirmed that serum IgA1 in Japanese IgAN patients is also highly reactive with this lectin (Figure 5a) [32,33]. However, there are some limitations to the lectin-based assay. Lectin activity and stability may vary from lot to lot [152]. Additionally, lectin binding to GalNAc is affected by sialylation of GalNAc and Gal in the clustered IgA1 *O*-glycans [153].

For a lectin-independent detection of Gd-IgA1, monoclonal antibodies against human Gd-IgA1 HR peptides were developed, such as KM55 and 35A12, and ELISA methods using these monoclonal antibodies were established (Figure 5b) [152,154].

Although assays using lectin and monoclonal antibodies can detect Gd-IgA1, these methods cannot determine the number and the attachment site of *O*-glycans. The specific changes in IgA1 HR *O*-glycoforms in IgAN patients cannot be assessed by these analytical methods. For a detailed analysis of the *O*-glycoforms of IgA1 HR, which harbors clustered *O*-glycans, the HR glycopeptides need to be analyzed using high-resolution mass spectrometry combined with nanoflow liquid chromatography after digestion with endopeptidases, such as trypsin (Figure 5c). To identify the sites of *O*-glycan attachment, electron transfer dissociation (ETD) tandem MS has been used [52,53]. We previously developed an automated quantitative analytical workflow for profiling HR *O*-glycopeptide and for the detection and quantification of galactose-deficient glycan attachment sites [55]. Using this analytical method, the most frequently utilized Gal-deficient site in serum IgA1 in healthy subjects was T^236^, followed by S^230^, T^233^, T^228^, and S^232^ [55].

### 4.2. Reliability of Gd-IgA1 as a Diagnostic and Prognostic Biomarker

Serum levels of Gd-IgA1 detected by lectin ELISA are significantly higher in patients with IgAN than in healthy controls or patients with other renal diseases [33,155]. However, it is still controversial whether higher levels of Gd-IgA1 recognized by HAA lectin are associated with clinical severity and outcome [32,33,155,156]. A couple of studies showed the predictive value of Gd-IgA1 serum levels [157,158]. Longitudinal changes in Gd-IgA1 serum levels before and after combination therapy, palatine tonsillectomy and steroid pulse therapy do not provide a clear picture [33,156]. Furthermore, rituximab therapy does not alter the Gd-IgA1 levels obtained by HAA ELISA [159].

Recently, several reports have been published on the validation of Gd-IgA1 measured using KM55. The serum level of Gd-IgA1 measured with KM55 is higher in patients with IgAN and HSPN than in patients with lupus nephritis (LN), ANCA-associated vasculitis (AAV), and minimal change disease in a Japanese cohort [160]. In addition, KM55 recognizes glomerular IgA1 in patients with IgAN and IgA vasculitis with nephritis (IgA-VN) [161]. Furthermore, elevation of serum KM55 Gd-IgA1 levels is associated with histopathologically advanced IgAN and predicts post-transplant recurrent IgAN [160,162]. KM55 recognizes GalNAc residues on Thr^225^ and Thr^233^ in synthetic glycopeptides in the PST(GalNAc)PP motif in IgA1 HR [163], but it is not known whether IgA1 with Gal-deficient *O*-glycan in these specific sites is associated with the pathogenesis of IgAN and IgA-VN.

However, the KM55-based method showed that serum Gd-IgA1 levels are elevated in patients with secondary IgAN to the same degree as in those with primary IgAN [164]. Immunostaining of Gd-IgA1 by KM55 was also observed in secondary IgAN and other IgA-depositing diseases, such as IgAN with hepatitis B virus, LN, cirrhosis, Crohn’s disease, *Staphylococcus*-associated glomerulonephritis, and psoriasis [165,166]. Thus, the reliability of Gd-IgA1 measured by KM55 should be evaluated in a larger multicenter cohort.

## 5. Conclusions

The elevation in serum Gd-IgA1 levels and immunodeposits enriched for Gd-IgA1 glycoforms are characteristic features of IgAN. To clarify the characteristics of disease-specific IgA1, many studies have attempted to use lectin assays, monoclonal antibodies, mass spectrometry, and genetic and other types of analyses. However, the detailed characterization of *O*-glycoforms of IgA1 associated with IgAN is still to be performed. In addition, the effect of different *O*-glycoforms in IgA1 HR on the formation of immune complexes, mesangial deposition, and glomerular injury has not been fully elucidated. Table 1 summarizes what we know and what we do not know about glycosylation of IgA1 in IgAN.

Recent genetic studies have shown that *O*-glycosylation of IgA1 is influenced by genetic variants of *O*-glycosyltransferase, and racial differences in variants have also been noted. Therefore, racial factors should be considered when investigating disease-specific *O*-glycoforms. By clarifying the disease-specific *O*-glycoform(s), the mechanism underlying aberrant glycosylation, and the effects of glycosylation changes, it is hoped that the elucidation of the etiology and development of diagnostic markers and new therapies for this disease will be advanced.

## Figures and Tables

**Figure 1 jcm-10-03467-f001:**
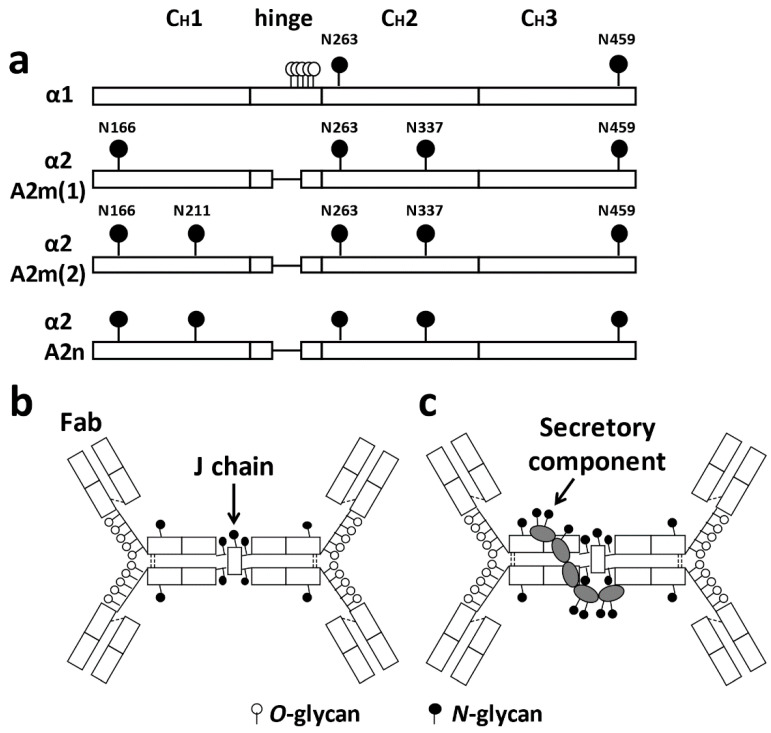
Molecular structure of IgA and its glycosylation sites. (**a**) Human IgA has two subclasses: IgA1 and IgA2. IgA1 harbors clustered *O*-glycans in its hinge region. IgA1 and IgA2 have several *N*-glycans in their constant region of heavy chains. IgA1 has two *N*-glycosylation sites at asparagine (Asn)^263^ and Asn^459^. Three allotypes of IgA2 are known, designated A2m (1), A2m (2), and IgA2 (n). All allotypes have *N*-glycans at Asn^166^, Asn^263^, Asn^337^, and Asn^459^. A2m (2) and IgA2 (n) allotypes have a fifth *N*-glycan at Asn^211^ [45]. (**b**,**c**) Schematic representation of dimeric IgA1 and secretory IgA1. Both the joining chain (J chain) and the secretory component have *N*-glycan(s). Fab, antigen-binding fragment.

**Figure 2 jcm-10-03467-f002:**
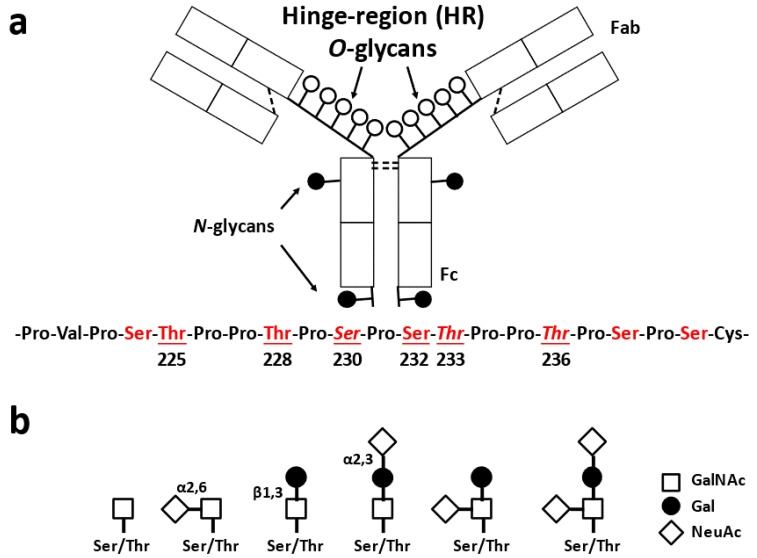
Schematic representation of human IgA1. The IgA1 heavy chain has three to six *O*-glycans in its hinge region (HR) and two *N*-glycosylation sites [46,54]. There are nine potential *O*-glycosylation sites, marked in red font, of which up to six sites can be *O*-glycosylated (underlined serine (Ser) and threonine (Thr)). Ser/Thr in italic (230, 233, 236) show the frequent sites with galactose (Gal)-deficient *O*-glycan (**a**) [50,52,53,55]. There are *O*-glycan variants of circulatory IgA1. *N*-acetylgalactosamine (GalNAc) is attached to Ser/Thr residues and can be extended by the attachment of Gal to GalNAc residues. GalNAc or Gal or both can be sialylated. Due of diversity of the glycan attachment sites, the number of *O*-glycans in HR, and variability of *O*-glycan structures, IgA1-HR *O*-glycoforms exhibit wide heterogeneity. Gd-IgA1-specific antibodies are considered to recognize Gal-deficient IgA1 glycoforms with terminal GalNAc (left structure). NeuAc, *N*-acetylneuraminic acid (**b**).

**Figure 3 jcm-10-03467-f003:**
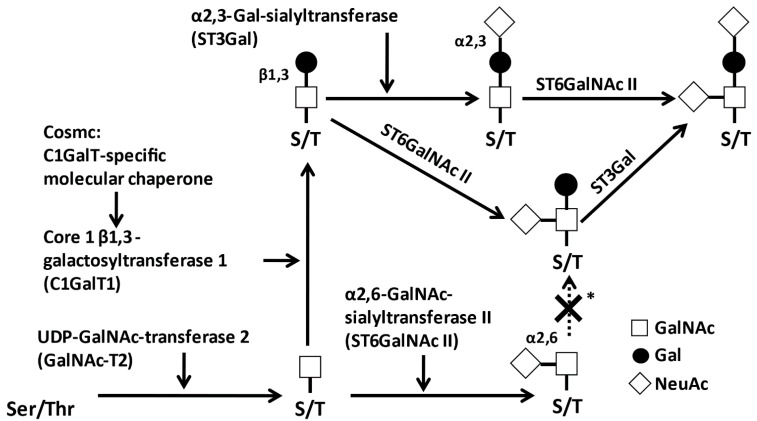
*O*-Glycosylation pathways of the human IgA1 hinge region (HR). The biosynthesis process starts with the attachment of *N*-acetylgalactosamine (GalNAc) to Ser/Thr residues in the HR by UDP-GalNAc-transferase 2 (GalNAc-T2). GalNAc residues are extended by galactose or *N*-acetylneuraminic acid (NeuAc) by core 1 β1,3-galactosyltransferase (C1GalT1) and its molecular chaperone Cosmc or α2,6 sialyltransferase (ST6GalNAc2), respectively. Finally, the *O*-glycan structure is completed by attachment of NeuAc to the galactose residue and/or GalNAc residues, each of which is mediated by α2,3-sialyltransferase (ST3Gal1) and ST6GalNAc2. The sialylation of GalNAc before the attachment of galactose prevents galactosylation of GalNAc (marked by *) [62].

**Figure 4 jcm-10-03467-f004:**
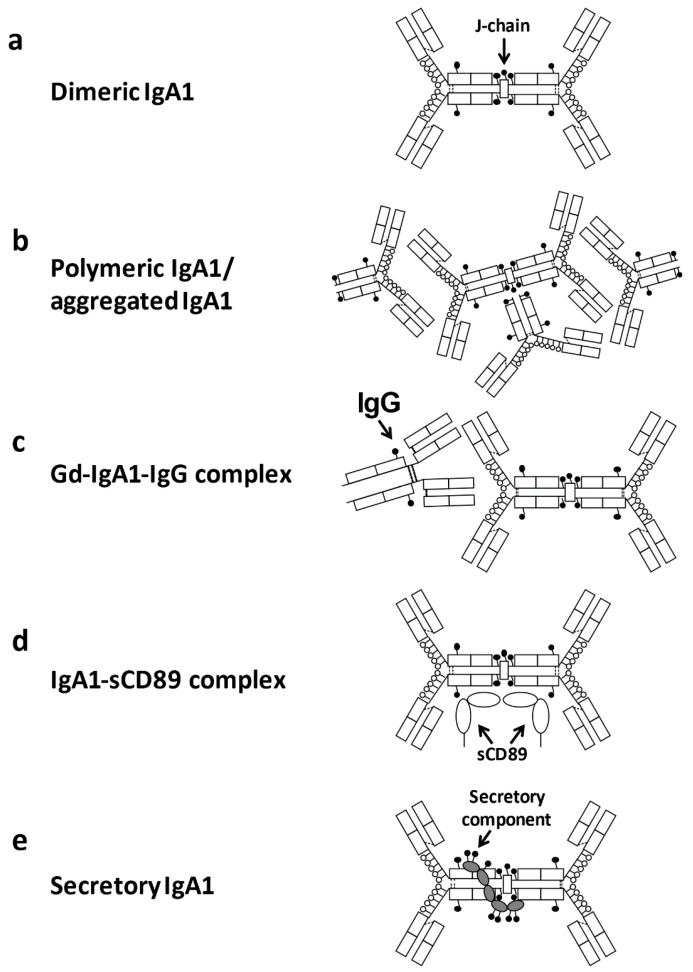
Macromolecular forms of IgA1. Although mesangially deposited IgA1 has not been fully characterized, Gd-IgA1-IgG (or IgA) immune complexes, IgA-IgA receptor complexes, self-aggregated IgA1 proteins, other serum protein complexes with IgA1, and secretory IgA1 are possible forms in the kidney deposits [23]. Dimeric IgA1 is composed of two monomeric IgA1 connected by joining chain (J chain) (**a**). Larger molecular forms of IgA1 may include complexes/aggregates of dimeric IgA1 and monomeric IgA1. Aberrantly glycosylated IgA1 may be prone to aggregation [127]. (**b**). Incomplete galactosylation of *O*-glycans in the IgA1 hinge region results in the exposure of terminal GalNAc and is recognized by autoantibodies (IgG of IgA), leading to the formation of IgA1-containing immune complexes (**c**). IgA1 complexes with soluble CD89 (sCD89) may be formed from CD89 cleaved from the surface of monocytes/macrophages (**d**). Secretory IgA1 consists of dimeric IgA1 with J chain and the secretory component (**e**). These macromolecular IgA1 forms (**a**–**e**) can form complexes with other serum proteins.

**Figure 5 jcm-10-03467-f005:**
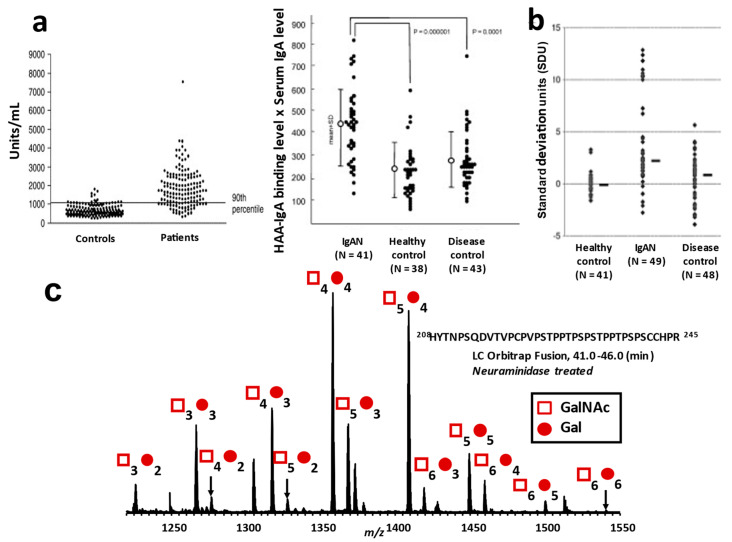
Detection of galactose-deficient IgA1 (Gd-IgA1). Serum Gd-IgA1 levels measured by *Helix aspersa* agglutinin (HAA)-based ELISA, due to HAA specificity for terminal GalNAc, is significantly higher in patients with IgAN than in healthy subjects (**a**) [32,33]. The left figure of (**a**) was published in *Kidney International* 2007, 71, 1148-54, Moldoveanu, Z. et al., Copyright 2007 Elsevier Inc and is republished with permission. The right figure of (**a**) is republished with permission of Oxford University Press, from *Nephrology Dialysis Transplantation* 2008, 23, 1931-9, Shimozato, S. et al., Copyright 2008 Oxford University Press. Detection of Gd-IgA1 using monoclonal antibody against human Gd-IgA1 hinge region (HR) peptide (**b**) [154]. The figure (**b**) was published in *Journal of Nephrology* 2015, 28, 181-6, Hiki, Y. et al., Copyright 2014, The Author(s). This article is under the terms of the Creative Commons CC BY license. A variety of IgA1 HR *O*-glycoforms can be detected by high-resolution mass spectrometry according to the difference in mass arising from the number of attached monosaccharides to the amino acid backbone of the IgA1 HR (His^208^-Arg^245^). The number of *N*-acetylgalactosamine (GalNAc; □) and galactose (Gal; ●) are shown above the individual peaks (**c**).

**Table 1 jcm-10-03467-t001:** What we know and what we do not know about *O*-glycosylation of IgA1 in IgAN.

**Characteristics of serum IgA1 in IgAN**		
	High reactivity with lectin from *Helix aspersa* (HAA)	Moldoveanu,2007 [32], Shimozato, 2008 [33]
	High reactivity with monoclonal antibodies	Yasutake, 2015 [152], Hiki, 2015 [154]
**IgA1 carbohydrates in IgAN assessed by mass spectrometry**		
	Decrease of galactose (Gal)	Hiki, 1998 [167], Inoue, 2012 [168], Nakazawa, 2019 [169]
	Decrease of *N*-acetylgalactosamine (GalNAc)	Hiki,1998 [167], Odani, 2000 [170], Inoue, 2012 [168], Nakazawa, 2019 [169]
	Decrease of sialic acid	Odani, 2000 [170]
**Sites of glycan attachment**		
	Occur in specific sites	Renfrow, 2005 [50], Iwasaki, 2003 [51], Takahashi, 2010 [52], Takahashi, 2012 [53]
	Disease in specific sites	Needs to be investigated
**Expression of *O*-glycosyltransferases in B cells**		
	No decrease in the ratio of *C1GALT1*: *GALNT2* or *C1GALT1*: *C1GALT1C1*	Buck, 2008 [171]
	Expression of *C1GALT1* and *ST6GALNAC2* altered in IgA1-producing cell lines from IgAN patients	Suzuki, 2008 [76]
	Decreased *C1GALT1* expression in IgAN CD19+ B cells	Xing, 2020 [172]
***O*-Glycosylation characteristics of polymeric IgA1**		
	Polymeric IgA1 shows higher reactivity with HAA than in monomeric IgA1	Oortwijin, 2006 [149], Suzuki, 2008 [76], Novak, 2011 [173]
	Polymeric IgA1 interacted with CD71 and its interaction is enhanced by sialidase and β-galactosidase	Moura, 2004 [139]
**Epitopes recognized by IgG/A autoantibodies**		
		Needs to be investigated
**Clinical factors related to glycosylation of IgA1**		
Genetic influences	Gd-IgA1 levels are highly inherited and affected by variants of glycosyltransferase	Tam,2009 [82], Kiryluk, 2011 [83], Gharavi, 2008 [79], Lin, 2009 [84], Hastings, 2010 [85], Gale, 2017 [95], Kiryluk, 2017 [96]
	Susceptibility to IgAN is affected by variants of glycosyltransferase	Li 2007 [86,89], Pirulli, 2009 [87], Zhu, 2009, [88]
Race differences	Gd-IgA1 levels elevated in Caucasian patients	Gale,2017 [95]
	Gd-IgA1-increasing allele is common in Europeans	Kiryluk, 2017 [96]
	MicroRNA regulating *C1GALT1* and *GALNT2* overexpressed in Caucasian patients	Serino, 2016 [100]
Age differences		Needs to be investigated
Disease activity	No association of Gd-IgA1 levels measured by HAA-based ELISA with disease activity	Moldoveanu, 2007 [32], Shimozato, 2008 [33]
	Positive association of Gd-IgA1 level measured by HAA-based ELISA with disease activity	Suzuki, 2014 [156], Sun, 2016 [155], Zhao, 2012 [157], Maixnerova, 2019 [158]
	Gd-IgA1 levels measured by KM55 associated with disease progression or recurrence	Wada, 2018 [160], Temurhan, 2017 [162]
Longitudinal changes	Longitudinal changes of Gd-IgA1 serum levels by HAA ELISA before and after therapy need additional studies	Shimozato, 2008 [33], Suzuki, 2014 [156], Lafayette, 2017 [159]
	*O*-Glycoform analyzed by MS changes between, before and after therapy	Iwatani, 2012 [174]

## Data Availability

No new data were created or analyzed in this study. Data sharing is not applicable to this article.

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
