# Peer review of "Aberrantly Glycosylated IgA1 in IgA Nephropathy: What We Know and What We Don’t Know"

_jcm, 2021, doi:10.3390/jcm10163467_

Round 1

Reviewer 1 Report

My comments are as follows:

Page 2/25, line 76. Is there any other more solid proof of complement activation besides the C3 staining in tissue? That should be mentioned here. Because C3 staining in tissue can be highly non-specific and can be seen in any form of glomerular injury. Conversely, absence of C3 staining does not exclude complement-mediated injury.

Can the authors shed some light on a subset of IgA nephropathy cases which present in young patients with kidney biopsy showing already chronic renal injury almost ESRD. Any studies done on those kind of cases?

Otherwise, overall, the review is well-written. It is a good recapitulation of the existing literature .

Author Response

We thank the reviewer for their suggestions and comments. Below, we are providing point-by-point responses.

  • Page 2/25, line 76. Is there any other more solid proof of complement activation besides the C3 staining in tissue? That should be mentioned here. Because C3 staining in tissue can be highly non-specific and can be seen in any form of glomerular injury. Conversely, absence of C3 staining does not exclude complement-mediated injury.

Response: Thank you for your kind comments. Besides the C3 staining in tissue, presence of C3 in the circulating IgA1-containing immune complexes of patients with IgAN have been reported. We added the sentence “as well as by the presence of C3 in the circulating IgA1-containing immune complexes” in line 76, and some references [37-42].

  • Can the authors shed some light on a subset of IgA nephropathy cases which present in young patients with kidney biopsy showing already chronic renal injury almost ESRD. Any studies done on those kind of cases?

Response: Thank you for your kind suggestions. To the best of our knowledge and literature search, there are no studies on the pathological involvement or clinical significance of Gd-IgA1 in progressive IgAN in children. It would be important to measure Gd-IgA1 and longitudinal changes of Gd-IgA1 in such cases with young onset and rapid progression.

Reviewer 2 Report

The authors provide a review article primarily focused on the pathogenesis of IgA nephropathy. While the content of this manuscript is well chosen, the article must be extensively revised for readability before publication, perhaps by one of the authors in the United States.

I suggest no major changes in terms of the content. However, there is a need for substantial revision due to the volume of minor changes. The following is a non-exhaustive list of minor revisions that I would make for readability just in the first 4 paragraphs of the manuscript. I would also suggest re-organization of the introductory sections, as the message is not clear.

  1. Line 29. I would replace “Immunodeposits” with “deposition”.
  2. Line 31. I would write “Recurrence of IgA nephropathy is common in patients after transplantation”
  3. Line 44. I would replace “IgAN” with “Patients with IgAN.
  4. Line 44-51, especially 49. It is not clear what is meant by the author here.
  5. Line 55. I would remove “directly” as qualification is already made afterwards.
  6. Line 55 & 56. I would not use the term “Co-deposits”, but would say “it is accompanied by deposition of complements”.
  7. Line 59. I would remove “of disease” and replace “concurrent” by “concurrently”.
  8. Line 64. I would replace “firstly” with “first”.

Author Response

We thank the reviewer for their suggestions and comments. Below, we are providing point-by-point responses.

The authors provide a review article primarily focused on the pathogenesis of IgA nephropathy. While the content of this manuscript is well chosen, the article must be extensively revised for readability before publication, perhaps by one of the authors in the United States.

I suggest no major changes in terms of the content. However, there is a need for substantial revision due to the volume of minor changes. The following is a non-exhaustive list of minor revisions that I would make for readability just in the first 4 paragraphs of the manuscript. I would also suggest re-organization of the introductory sections, as the message is not clear.

Response: Thank you for your kind suggestions. The article was revised by the authors in the United States and the introductory sections were reorganized to make the message clear. We also revised the following list of minor revisions.

  • Line 29. I would replace “Immunodeposits” with “deposition”.

Response: We replaced “Immunodeposits” with “deposition”.

  • Line 31. I would write “Recurrence of IgA nephropathy is common in patients after transplantation”

Response: We revised the sentence.

  • Line 44. I would replace “IgAN” with “Patients with IgAN.

Response: We replaced “IgAN” with “Patients with IgAN.

  • Line 44-51, especially 49. It is not clear what is meant by the author here.

Response: We revised this paragraph as below.

“Patients with IgAN present with a variety of clinical, laboratory, and histopathological findings. In addition to primary IgAN, many patients with liver disease, collagen disease, inflammatory bowel syndrome, malignant tumor, and infection exhibit IgA glomerular deposition, i.e., secondary IgAN [15,16]. Due to these many clinical features and types of secondary IgAN, it has been assumed that IgAN may represent a group of diseases rather than a single disease [17,18].”

  • Line 55. I would remove “directly” as qualification is already made afterwards.

Response: We removed the word “directly”.

  • Line 55 & 56. I would not use the term “Co-deposits”, but would say “it is accompanied by deposition of complements”.

Response: We We replaced “complement co-deposits” with “deposition of complement”.

  • Line 59. I would remove “of disease” and replace “concurrent” by “concurrently”.

Response: We removed “of disease” and replace “concurrent” by “concurrently”

  • Line 64. I would replace “firstly” with “first”.

Response: We replaced “firstly” with “first”.

Reviewer 3 Report

complete and well-documented review of what is known and remains to be known about gd-IgA during N-IgA. It should be emphasized that the proof of the pathogenic role of gd-IgA has not been definitively demonstrated.

minor: line 221   and. A       ,

Author Response

We thank the reviewer for their suggestions and comments. Below, we are providing point-by-point responses.

  • complete and well-documented review of what is known and remains to be known about gd-IgA during N-IgA. It should be emphasized that the proof of the pathogenic role of gd-IgA has not been definitively demonstrated.

Response: Thank you for your comments. Importantly, as the reviewer pointed out, serum level of Gd-IgA1 is elevated in some of IgAN patients, but its pathogenic role is not well understood. Therefore, in this review, we try to summarize what we currently know and what we do not know about the pathogenicity of aberrantly glycosylated IgA1 including Gd-IgA1 in IgAN. We hope that this comprehensive review will lead to further discussion on the role of glycosylation of IgA1 in IgAN. As mentioned in the paragraph of conclusion, we believe that further studies are needed to elucidate its true pathogenicity, and one of them should include structural analysis of mesangial deposited IgA1.

  • minor: line 221   and. A       ,

Response: Thank you. We deleted “and”.

Round 2

Reviewer 2 Report

The manuscript is improved. There are still places with awkward phrasing (especially last page), but this is minor. In general, I suggest avoiding passive voice to make the article clearer.